# Extracellular-Matrix-Mimetic Hydrogels by Using Nanomaterials

**DOI:** 10.3390/ijms26114987

**Published:** 2025-05-22

**Authors:** Do Gyun Kim, Gi Doo Cha

**Affiliations:** Department of Systems Biotechnology, Chung-Ang University, Anseong 17546, Republic of Korea; kdg4376@cau.ac.kr

**Keywords:** extracellular matrix, nanomaterial, synthetic ECM-mimetic hydrogel, ECM-mimetic hydrogel, 3D cell culture, tissue engineering

## Abstract

Matrigel, a tumor-derived basement membrane extract, has been commercially used in the field of cell culture and tissue engineering due to its extracellular-matrix-mimetic (ECM-mimetic) properties. However, its batch-to-batch variability and limited mechanical tunability hinder reproducibility and clinical translation. To overcome these issues, synthetic ECM-mimetic hydrogels have been developed to improve reproducibility and biocompatibility. While they are effective in mimicking ECMs, these materials must go beyond passive replication by implementing the complex functionalities of the ECM. The integration of nanomaterials with hydrogel could address this need by reinforcing mechanical properties, enabling various functionalities, and featuring dynamic responsiveness. In this review, we present the evolution from Matrigel to ECM-mimetic hydrogels and ECM-mimetic hydrogel nanocomposites, exploring their key advancements and challenges. We will discuss the advantages and disadvantages of the transition from Matrigel to ECM-mimetic hydrogels and ECM-mimetic hydrogel nanocomposites, along with cases that have addressed Matrigel’s limitations and added new functionalities. Furthermore, we discuss future directions for the design of the ECM-mimetic hydrogels, emphasizing how nanotechnology strategies can drive innovation in tissue engineering and regenerative medicine.

## 1. Introduction

The extracellular matrix (ECM) functions to support cell attachment, proliferation, differentiation, and migration; the ECM is actively explored in the field of cell culture and tissue engineering [1,2,3]. ECMs in living bodies consist of a very complex structure and feature various functionalities; therefore, research into ways of mimicking the ECM is in demand, as it can aid in the creation of biomimetic scaffolds with high performance [3,4]. Matrigel, a tumor-derived basement membrane extract, has been extensively used for these purposes, due to its ability to provide ECM-like biochemical and structural cues [5,6,7]. Matrigel primarily consists of laminin, collagen IV, and entactin, forming a gelatinous structure that mimics the native basement membrane [8,9]. These components provide an environment conducive to cell adhesion and differentiation, making Matrigel suitable for stem cell culture and organoid formation [10,11,12].

Despite the widespread use of Matrigel, there are significant limitations in its reproducibility, mechanical strength, and suitability for clinical translation. Batch-to-batch variability and undefined composition contribute to inconsistencies in the biochemical properties of Matrigel, leading to challenges in experimental reproducibility [9,13,14]. Additionally, its poor mechanical tunability limits its applicability in studies requiring precise control over matrix stiffness, making it unsuitable for modeling tissues with defined mechanical properties [4]. Furthermore, as a tumor-derived animal product, Matrigel raises ethical concerns and encounters potential risks of xenogeneic contamination, limiting its applications in clinical settings [13,14].

To overcome these limitations, researchers have sought to develop ECM-mimetic hydrogels that can replicate the key features of the ECM while offering improved reproducibility and mechanical stability. These hydrogels, composed of natural (e.g., hyaluronic acid, collagen, gelatin, alginate) or synthetic (e.g., polyethylene glycol and polyacrylamide) polymers, are designed to integrate biochemical cues (e.g., cell-adhesion motifs, enzymatic degradability, and growth factor binding) with tunable stiffness and viscoelasticity to match various tissue environments [15,16]. Additionally, its structure should ensure high reproducibility for practical applications in 3D (three-dimensional) bioprinting and tissue engineering. Despite these advancements, ECM-mimetic hydrogels often require additional modifications to optimize their mechanical robustness, bioactivity, and adaptability to dynamic environments [16,17]. Various strategies, such as combining synthetic and natural biomaterials into hybrid scaffolds, have been employed to enhance hydrogel performance, improving stability, endowing tunability, and balancing biocompatibility/durability [16,17,18,19].

However, these approaches are often insufficient for achieving the functional complexity required for advanced cell culturing and tissue engineering applications. The integration of nanomaterials into hydrogels represents a breakthrough in this field, significantly expanding their mechanical and functional properties [20,21,22,23,24,25,26]. Nanomaterials such as carbon nanotubes, gold nanoparticles, graphene, magnetic nanocomposites, and ceramic nanofillers have been incorporated into hydrogels to enhance their performance [18,19,22,23,24,25,27,28]. They lead to improved mechanical strength by reinforcing hydrogel matrices, increasing electrical conductivity for implementing electrical functions, and enhancing printability for realizing bioelectronics [18,19,22,23,24,25,27] and responsiveness to stimuli [26,28,29,30,31].

In this review, we explore ECM-mimetic hydrogel nanocomposites, emphasizing their advancements and addressing the limitations of Matrigel (Figure 1). We will also discuss the advantages and disadvantages of the evolution from Matrigel to ECM-mimetic hydrogel nanocomposites, as well as the respective cases where ECM-mimetic hydrogels have been used in overcoming the disadvantages of Matrigel and where ECM-mimetic hydrogel nanocomposites have additional functionality. Accordingly, the study aims to predict next-generation ECM-mimetic hydrogel nanocomposites that can enable precise control over cell behavior and tissue development, ultimately advancing fields such as regenerative medicine, soft bioelectronics, and organoid research.

## 2. The Extracellular Matrix and the Evolution of Biomimetic Hydrogels

The ECM is a highly dynamic and structurally complex network that plays a crucial role in regulating cell behavior, tissue organization, and biochemical signaling [1,2]. It consists of fibrous proteins (e.g., collagen, elastin), glycoproteins (e.g., fibronectin, laminin), and proteoglycans, which collectively provide mechanical stability, biochemical signaling, and a supportive microenvironment for cells [1,2]. In addition to its structural role, the ECM acts as a reservoir for bioactive molecules, regulating cell adhesion, migration, proliferation, and differentiation [32,33,34]. Given its importance in tissue homeostasis and regeneration, mimicking ECM properties has been a key challenge in cell culture, disease modeling, and tissue engineering [3,4].

Matrigel, a basement membrane extract derived from Engelbreth–Holm–Swarm (EHS) mouse sarcoma, has been widely used as a biologically active ECM-mimetic material [5,6,7]. Composed mainly of laminin (~60%), collagen IV (~30%), entactin (~8%), and heparan sulfate proteoglycans [8], Matrigel provides key biochemical and mechanical cues that are essential for cell culture, organoid formation, and tumor modeling [11,12,35,36].

However, Matrigel suffers from batch inconsistency, insufficient mechanical robustness, and lack of compositional specificity, leading to challenges in experimental reproducibility and clinical applications [13,14,37,38,39,40,41]. Furthermore, as a tumor-derived product originating from murine sarcoma, Matrigel contains undefined and variable growth factors that can unintentionally promote stem cell overproliferation or differentiation along undesirable lineages. Its low and inconsistent mechanical stiffness also impairs the delivery of precise mechanotransduction cues, which are essential for guiding cell fate decisions. In addition to these biological limitations, Matrigel presents ethical concerns and risks of xenogenic contamination, further restricting its translational potential [13,14,40,41]. To overcome these limitations, researchers have developed ECM-mimetic hydrogels, with improved reproducibility, tunability, and biocompatibility [42,43,44,45]. ECM-mimetic hydrogels are engineered using biocompatible polymers, which include natural (e.g., hyaluronic acid, alginate) or synthetic (e.g., polyethylene glycol (PEG)) polyacrylamide (PA) [15]. These materials are crosslinked to form hydrophilic and 3D polymeric networks that mimic ECM structures. Their adjustable mechanical and biochemical properties, by controlling precursor ratio and amount, allow for precise control over stiffness, degradation rates, and bioactive molecule release [44,45]. This advantage renders them more adaptable for customized cell culture and regenerative medicine applications. Furthermore, unlike Matrigel, ECM-mimetic hydrogels provide a defined composition, eliminating batch variability and the risk of xenogenic contamination [43]. However, ECM-mimetic hydrogels still face limitations such as complex synthesis or functionalization procedures and limited long-term stability, particularly in dynamic or in vivo environments [17].

While ECM-mimetic hydrogels have significantly advanced ECM mimicry, they primarily focus on structural and biochemical replication without incorporating additional functionalities. To further enhance bioactivity, responsiveness, and mechanical properties, researchers have developed hydrogels incorporating nanomaterials [15,18,46]. These functionalities can be broadly grouped into the following categories: electromagnetic responsiveness, exemplified by conductive or magnetically aligned systems; mechanical reinforcement, which enhances toughness and stretchability; and stimuli-responsiveness, which enables spatiotemporal control over cell behaviors such as migration and differentiation. These additional functionalities are often mediated through external triggers such as electric fields, magnetic actuation, or photothermal effects, thereby expanding their applicability in bioelectronics, 3D bioprinting, and dynamic tissue scaffolds [18,19,22,23,24,25]. Nanomaterial incorporation also allows hydrogels to respond to external cues, enabling precise control over cell fate, tissue remodeling, and organoid engineering [26,30]. Nevertheless, ECM-mimetic hydrogel nanocomposites are not without limitations; concerns over the potential cytotoxicity of nanomaterials and increased production costs present new challenges [47,48]. However, these limitations could be handled through material optimization and surface modification strategies, which could ensure the tailored performance and functionality of hydrogels, increasing their translation potential [49,50].

## 3. ECM-Mimetic Hydrogels: An Alternative Approach to Matrigel for ECM Mimicry

Synthetic ECM-mimetic hydrogels are indispensable platforms for replicating tissue-like environments in various biomedical applications [3,15,28,32,51]. Therefore, many researchers have focused on overcoming the disadvantages of Matrigel [43,52,53]. Such approaches can be classified into two categories: ECM-mimetic hydrogel and ECM-mimetic hydrogel nanocomposite [4,17]. Both approaches share the common goal of mimicking the native ECM but differ in their design strategies and functionalities [16].

The polymer backbone of the synthetic hydrogels is typically engineered to reproduce the structural and biochemical features of the ECM, often through the polymer network composition or the modification of functional moieties [15,32]. For example, key matrix components or bioactive motifs such as hyaluronic acid, collagen, or RGD peptides are engineered to be incorporated within the hydrogel network [54,55]. Their development has been largely guided by the need to overcome the limitations of Matrigel, offering improved reproducibility, defined composition, and tunable physical properties [43,56]. These hydrogels allow precise control over microenvironmental factors such as stiffness, biochemical signaling, and nutrient diffusion [2,56]. For example, incorporation of RGD peptides enhances cell–matrix adhesion [54], HA–collagen hybrids improve physiological relevance and compatibility [32], and gelatin-based networks offer adjustable permeability and mechanical flexibility [57].

On the other hand, ECM-mimetic hydrogels are formulated by integrating nanomaterials—such as carbon nanotubes, gold nanoparticles, or graphene derivatives—into the hydrogel network [4,58,59]. These systems aim to extend the functionality of conventional hydrogels by introducing new properties to nanomaterials, such as electromagnetic responsiveness, which are difficult to achieve through polymer engineering alone [4,17,27,28,29,31]. This incorporation enables us to expand their applicability in advanced bioengineering contexts.

Table 1 summarizes the representative examples across two ECM-mimetic hydrogel domains, along with their major applications in cell culture modeling and tissue engineering. Each example outlines the hydrogel’s material composition and highlights its functional advantages, serving as a comparative framework for understanding how different hydrogel designs contribute to specific biological outcomes. The following sections further explore these applications in detail, focusing on how both ECM-mimetic hydrogels and ECM-mimetic hydrogel nanocomposites can be tailored to meet diverse research in biomedical fields and clinical needs, with potential for commercialization.

## 4. Application of Synthetic ECM-Mimetic Hydrogels

Synthetic ECM-mimetic hydrogels can be broadly categorized into ECM-mimetic hydrogels and ECM-mimetic hydrogel nanocomposites. The design of ECM-mimetic hydrogels focuses on the hydrogel network for reproducing the ECM’s structural and biochemical cues; meanwhile, ECM-mimetic hydrogel nanocomposites incorporate functional nanomaterials to introduce additional responsiveness and mechanical enhancements. This section explores representative applications across two major domains—cell culture modeling and tissue engineering—demonstrating how material design improves biological behavior and function.

### 4.1. Cell Culture Modeling

Synthetic ECM-mimetic hydrogels offer a well-defined and reproducible platform for modeling vascular networks, addressing the major limitations associated with Matrigel. In a study by Lee et al. [21], a PEG hydrogel was engineered to present with controlled biochemical ligands and mechanical properties. The system is composed of independently tunable components (e.g., RGD cell adhesion peptides, VEGF-binding peptides, matrix metalloproteinase (MMP)-degradable crosslinkers, and PEG-based polymer backbones) that allow for the precise modulation of cell–matrix interactions, stiffness, and degradability (Figure 2A) [73,79,80]. A comparative analysis showed that the studied ECM-mimetic hydrogel supported superior endothelial network formation compared to those formed in Matrigel. Unlike Matrigel, which suffers from undefined composition and xenogeneic origin, the ECM-mimetic hydrogel system enables controlled presentation of adhesion motifs and matrix mechanics, leading to more standardized outcomes [80]. Moreover, the platform demonstrated broad applicability across multiple endothelial cell types, including both primary human umbilical vein endothelial cells (HUVECs) and induced pluripotent stem-cell-derived endothelial cells (iPSC-ECs), which supports its versatility for various vascular models.

The dimensional structure of the cellular microenvironment plays a critical role in regulating cell behavior. Epithelial cells cultured in the 3D matrix showed more life-like physiological morphologies and gene expression patterns compared to flat 2D (two-dimensional) cultures, demonstrating the importance of ECM-like architecture for cell culture (Figure 2D) [82,83,84]. Building on this, the study evaluated a PEG-based ECM-mimetic hydrogel with a 3D network against Matrigel in supporting 3D acinar structure formation (Figure 2E). The PEG hydrogel, when designed with appropriate mechanical and biochemical features, successfully induced lumen formation and cellular polarization in the same way as Matrigel. This suggests that structural mimicry alone, even without complex biological extracts, can replicate the key aspects of ECM functionality (Figure 2F) [85,86]. These findings support the potential of ECM-mimetic hydrogels as Matrigel alternatives by leveraging well-defined architectures to guide tissue-specific cellular organization and behavior [87].

Synthetic ECM-mimetic hydrogels offer improved reproducibility and tunable biochemical and mechanical properties; however, their limited biological complexity and static architecture often fall short in completely recapitulating the dynamic and heterogeneous nature of the native ECMs [66,81]. Although transcriptomic analyses have shown gene-level similarities to Matrigel-based cultures, functional interactions between matrix signals and cell fate decisions remain a challenge [66,81]. To address these limitations, recent efforts have utilized ECM-mimetic hydrogel nanocomposites, which integrate bio-instructive nanomaterials to impart additional functionality, dynamic responsiveness, and higher-order control over cellular behavior.

To expand the functional capabilities of ECM-mimetic hydrogels, nanomaterial integration has emerged as a key strategy. To improve the electrical and mechanical performance of cardiac tissue scaffolds, Shin et al. [77] incorporated carbon nanotubes (CNTs) into gelatin methacrylate (GelMA) hydrogels. Carbon nanotubes provided the hydrogels with electromagnetic networks and enhanced mechanical strength, thereby improving cardiomyocyte alignment, electrical coupling, and synchronous beating behavior. Notably, the CNT-GelMA matrix supported electrical stimulation, lowering the excitation threshold and enabling synchronized contractility of cardiomyocytes under external electrical cues. This approach resulted in a conductive and stiffened matrix that mimics the fibrous nature of native cardiac ECM (Figure 3A). As a result, the CNT-GelMA hydrogel promoted the alignment and elongation of cardiomyocytes with well-organized sarcomeric structures (Figure 3B), in contrast to the disorganized growth seen on pristine GelMA hydrogels [88]. The engineered tissues also showed enhanced beating activity and improved connectivity (Figure 3C), highlighting the role of CNTs in supporting functional maturation. Overall, this ECM-mimetic hydrogel nanocomposite system demonstrates how conductive nanomaterials can introduce tissue-specific functionalities and support the development of contractile and responsive characteristics in cardiac tissues [27].

Tognato et al. [65] developed a magnetically responsive hydrogel composed of PNIPAAm and GelMA, embedded with aligned magnetic nanoparticles (Figure 3D). In this system, iron oxide nanoparticles introduced magnetic responsiveness and an anisotropic nanoarchitecture, enabling magnetically guided cell alignment and enhanced myogenic differentiation. Under an external magnetic field, the nanoparticles aligned during gelation, producing an anisotropic matrix capable of translating magnetic stimuli into directional topographical cues. The anisotropic alignment of magnetic nanoparticles, formed via magnetic-field-assisted gelation, facilitated directional cell migration and elongation along the alignment axis (Figure 3E). Stem cells cultured within the aligned matrix exhibited enhanced myogenic differentiation, suggesting that topographical cues arising from the aligned structure—rather than direct magnetic stimulation—played a primary role in lineage commitment (Figure 3F), indicating that both structural cues and dynamic responsiveness contribute to lineage commitment. This platform illustrates the potential of stimuli-responsive ECM-mimetic hydrogel nanocomposite for guiding cellular behavior and differentiation, with promising applications in biomaterials fabrication and magnetically actuated soft robotics.

Although ECM-mimetic hydrogel nanocomposites equipped with conductive or magnetic nanomaterials offer new functionalities—such as anisotropic alignment, electrical signal conduction, and stimuli-responsiveness—their effects on cell fate remain controversial. In many cases, cellular outcomes are interpreted in relation to physical properties such as conductivity or anisotropy; yet, the direct links between these engineered functionalities and biological maturation processes are not clearly established [65,77]. Confounding factors such as protein polarization, indirect mechanical cues, or complex fabrication methods may influence results, complicating the interpretation of how external fields or embedded nanomaterials precisely modulate cell behavior and its resultant cytotoxicity.

### 4.2. Tissue Engineering

Although synthetic and ECM-mimetic hydrogel nanocomposites have shown great promise in modeling cellular behavior in vitro, there has been substantial research interest on extending their applicability for tissue engineering. Such hydrogels serve not only as scaffolds but also as functional regulators of tissue formation, integration, and regeneration. In this section, we highlight examples of hydrogel-based systems used in the engineering of bone, cardiac, vascular, and skin tissues, emphasizing how structural design, material functionality, and nanomaterial incorporation contribute to tissue-specific outcomes.

In a study by Wang et al., an ECM-mimetic hydrogel (LZM-SC/SS) was designed to modulate macrophage behavior through structural dynamics and bioactive adhesion motifs (Figure 4A,B) [89,90,91]. The hydrogel is formed via an amidation reaction between lysozyme (LZM)—providing a DGR tripeptide for cell adhesion—and 4-arm PEGs, where PEG-SS imparts hydrolytic degradability and PEG-SC ensures mechanical stability. This hybrid network not only mimics the dynamic remodeling of the native ECM but also enables spatial control over macrophage infiltration and M2 polarization, establishing a pro-regenerative immune microenvironment [90,91]. Subcutaneous injection of the LZM-SC/SS hydrogel in the LZM-SC/SS hydrogel in rats (Figure 4C,D) revealed robust cell infiltration and vascularization, in contrast to the control group [91]. Histological and immunofluorescent analysis confirmed enhanced M2 macrophage presence (CD206⁺) and angiogenesis (CD31⁺) within the dynamic hydrogel [91]. These results demonstrate the idea that matrix-intrinsic properties, rather than exogenous cytokines, can effectively guide immune responses and tissue regeneration. This work highlights the potential of structurally tunable and immune-instructive ECM-mimetic hydrogels for tissue engineering scaffolds [28,51].

To mimic the hierarchical structure of native skin, a 3D-bioprinted and bilayer hydrogel scaffold composed of GelMA, gelatin, and amniotic membrane extract (AME) was designed (Figure 4E) [94]. Keratinocytes and HUVECs were separately organized within epidermal and dermal compartments, replicating the layered architecture of human tissue. The hydrogel exhibited tunable mechanical properties through the modulation of GelMA and gelatin ratios, achieving elasticity and strength suitable for skin implantation (Figure 4F) [94,95]. Additionally, the incorporation of AME significantly enhanced angiogenic potential of the hydrogel, increasing VEGF expression and promoting tube-like vascular structures in HUVECs (Figure 4G) [94,95,96,97]. This approach demonstrates how ECM-mimetic hydrogel, combined with 3D bioprinting, can enable spatial control and vascularization in engineered skin tissues.

While ECM-mimetic hydrogels without nanomaterials have demonstrated regenerative potential through immunomodulation and vascularization, their effects are often supported by correlative findings rather than clearly defined causal mechanisms. Additionally, the absence of functional nanomaterials limits their ability to dynamically respond to external stimuli or precisely control microscale physical and biochemical cues. These limitations have motivated the development of nanocomposite hydrogels that offer enhanced responsiveness and fine-tuned regulation of cell–matrix interactions in tissue engineering contexts.

Nagahama et al. [98] introduced an injectable ECM-mimetic hydrogel nanocomposite combining PLGA-PEG-PLGA and LAPONITE^®^ to implement a self-replenishing ECM microenvironment for tissue regeneration [99,100]. Rather than functioning as a passive filler, the nanocomposite acts as a dynamic platform that gradually incorporates host-derived ECM molecules in situ, enabling long-term tissue support [29,51]. This unique functionality arises from the synergistic effects between PLGA-PEG-PLGA degradation and the high adsorption capacity of LAPONITE^®^. As the hydrogel degrades, LAPONITE^®^’s dual surface charges facilitate the retention of both positively and negatively charged ECM components such as collagen and heparin (Figure 5A) [100]. This leads to continuous remodeling of the hydrogel into the ECM-rich scaffold over time, without the need for exogenous supplementation. In a skeletal muscle injury model in rodents, the ECM-mimetic hydrogel nanocomposite significantly improved muscle regeneration and functional recovery, compared to Matrigel or PLGA-PEG-PLGA alone, as evidenced by increased muscle force over time (Figure 5B,C) [101,102]. These results validate the scaffold’s ability to autonomously engineer a regenerative microenvironment, implying its promising potential for minimally invasive scaffold.

Self-assembling peptide hydrogel based on arginine–alanine–aspartic acid–alanine (RADA) was suggested to offer a chemically defined bioactive matrix for 3D neural tissue engineering [105,106]. These peptides spontaneously form nanofibrous networks under physiological conditions, closely resembling the architecture of the native ECM (Figure 5D). The nanofiber hydrogel supported the encapsulation and differentiation of adult neural stem cells for over five months, surpassing the longevity and stability of conventional matrices. Specific bioactive motifs especially—such as SKPPGTSS, PFSSTKT, and RGD—could be incorporated into the peptide backbone to selectively influence cell fate, promoting controlled differentiation into neurons, astrocytes, or oligodendrocytes (Figure 5E). Without nanomaterial integration, the nanostructured fibrillar assembly itself functioned as a bio-instructive platform [107,108,109]. The peptide hydrogel significantly outperformed Matrigel and collagen I in terms of maintaining cell viability and promoting balanced neural differentiation over extended periods (Figure 5F). This work demonstrates that peptide nanofiber hydrogels can serve as powerful and modular ECM mimetics with high definition, functional adaptability, and stability for neural tissue engineering [107].

An ECM-mimetic hydrogel nanocomposite combining cellulose nanofibrils (CNFs) and low-concentration GelMA was developed to enhance structural stability, printability, and cell compatibility (Figure 5G) [110,111]. Electrostatic interactions between CNF and GelMA enabled effective UV crosslinking at a GelMA hydrogel concentration of only 0.2–1%, compared to the typically required standard (≥2.5%; Figure 5H) [110]. The incorporation of CNF also improved printing resolution and shape fidelity by optimizing the hydrogel’s viscoelastic properties, reducing nozzle clogging and deformation during extrusion (Figure 5G–J). Additionally, the CNF/GelMA scaffold supported high cell viability and proliferation, aided by the ECM-like structure and RGD-containing GelMA. SEM analysis revealed the micropore structure (5–20 μm) on the surface of the scaffold, which facilitates nutrient diffusion and cell infiltration (Figure 5J) [112,113]. The system highlights how nanocellulose can functionally reinforce hydrogels for high-resolution bioprinting and soft tissue regeneration.

Nanocomposite ECM-mimetic hydrogels have demonstrated advanced functionality in tissue engineering applications by incorporating nanomaterials that provide improved mechanical reinforcement, bioactive signal presentation, and structural fidelity [93,98,103,114]. These enhancements have enabled better ECM retention, increased cell viability, enhanced scaffold printability, and promoted tissue-specific differentiation and regeneration. However, despite these benefits, several challenges remain. In many cases, the added functionality is demonstrated over limited time frames or evaluated primarily through morphological and histological outcomes, without mechanistic clarification at the molecular level [98,103,104]. Furthermore, the individual contributions of each component within multi-phase hydrogel systems are not always clearly delineated, making it difficult to attribute specific regenerative outcomes to distinct material features. These limitations emphasize the need for more systematic studies that integrate long-term in vivo performance with molecular and functional validation, ensuring that nanocomposite hydrogels can reliably support tissue regeneration in clinical contexts.

## 5. Conclusions

The field of ECM-mimetic hydrogels has evolved rapidly, moving beyond biologically derived matrices like Matrigel toward ECM-mimetic hydrogel systems with defined compositions and tunable properties. In this review, we highlighted representative examples of ECM-mimetic hydrogels and ECM-mimetic hydrogel nanocomposites, applied to both cell culture modeling and tissue engineering, emphasizing their potential to recapitulate the structural and functional complexities of native ECMs. While ECM-mimetic hydrogel addressed key limitations of natural matrices—such as reproducibility and compositional control—the incorporation of nanomaterials has paved the way to a new horizon in ECM-mimetic hydrogels. By introducing functionalities like electromagnetic properties, mechanical reinforcement, and stimuli responsiveness, nanomaterials enable hydrogels to surpass the performance of passive ECM mimicry and actively engage in the regulation of cellular processes. Furthermore, nanomaterials can serve as platforms for targeted biomolecule delivery, structural anisotropy, or even microenvironment remodeling, offering unprecedented versatility.

However, despite these advancements, direct causal relationships between engineered functionalities and specific biological outcomes—such as lineage commitment, immune modulation, or functional tissue regeneration—are often insufficiently established. Additionally, concerns related to potential cytotoxicity, limited long-term biostability, and inconsistent performance under physiological conditions remain underexplored. Furthermore, few nanomaterials have received regulatory approval for clinical use; currently, no ECM-mimetic hydrogel nanocomposite has been approved for medical application, underscoring the need for thorough safety validation and translational standardization.

To realize an ideal ECM-mimetic hydrogel in the future, the rational integration of nanomaterials will be essential for engineering cell-instructive, responsive, and multifunctional matrices. Therefore, research should aim to focus on the safe synthesis of bioactive nanomaterials, scalable and reproducible fabrication techniques (such as high-resolution bioprinting), and long-term stability in vivo. Combining these efforts with molecular-level mechanistic validation and system-level biological analysis will be critical evidence in ensuring that functional improvements translate into robust therapeutic performances. By overcoming these challenges, ECM-mimetic hydrogel nanocomposites have the potential to innovate how we design biomimetic environments, rendering them not only structurally and biochemically accurate but also dynamically interactive, with cells for applications in regenerative medicine, organoid culture, and soft bioelectronics.

## Figures and Tables

**Figure 1 ijms-26-04987-f001:**
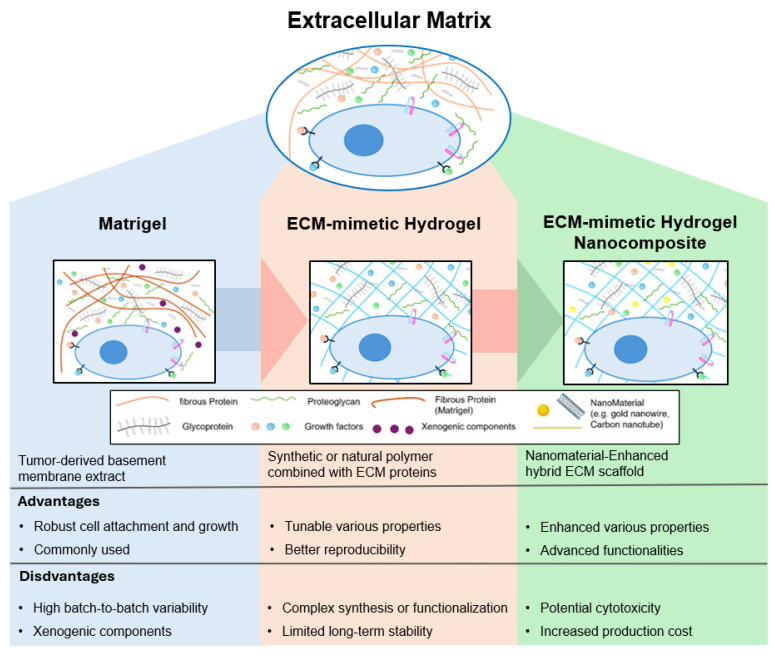
Evolution of synthetic ECM-mimetic hydrogel: the historical development of ECM-mimetic hydrogel from Matrigel to ECM-mimetic hydrogel and ECM-mimetic hydrogel nanocomposite. Each stage enhances reproducibility, tunability, and functionality, enabling more precise control over cell behavior and tissue development.

**Figure 2 ijms-26-04987-f002:**
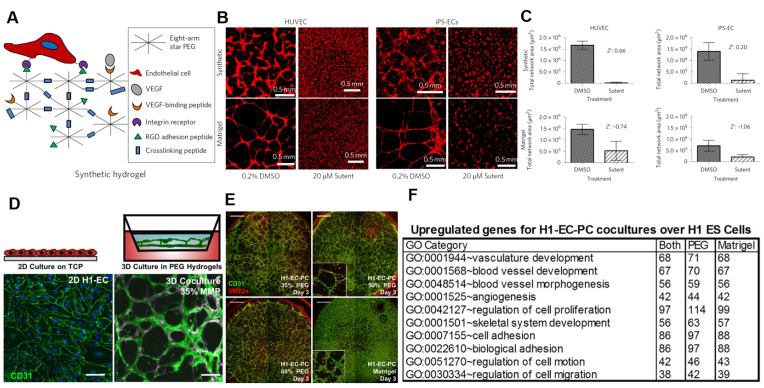
Applications of ECM-mimetic hydrogel on cell culture modeling. (**A**) Schematic of ECM-mimetic hydrogel design for ECM mimicry. (**B**) Images of HUVECs and iPSC-derived endothelial cells culturing from ECM-mimetic hydrogel and DMSO. (**C**) Comparison of assay quality between ECM-mimetic hydrogel and Matrigel using Z’ scores for endothelial network formation. Reproduced with permission from [81]. (**D**) Three-dimensional culture enhances epithelial cell phenotype, compared to 2D culture. Scale bar = 100 µm. (**E**) Acinar structures form in both PEG hydrogel and Matrigel. Scale bar = 1000 mm. (**F**) Genetical ECM mimicry including polarity and lumen formation is enabled without complex biological additives. Reproduced with permission from [66].

**Figure 3 ijms-26-04987-f003:**
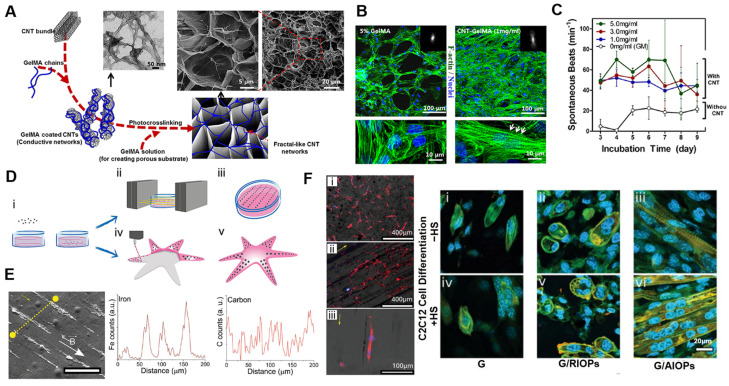
Applications of ECM-mimetic hydrogel nanocomposite for cell culture modeling. (**A**) CNTs enhance stiffness and conductivity that facilitates cell growth. (**B**) CNT-induced cues promote cardiomyocyte alignment and maturation, as evidenced by the elongated morphology of cardiac cells and the presence of well-developed F-actin cross-striations (white arrows, bottom right). (**C**) Engineered patches show enhanced beating behavior and electrical responsiveness. Reproduced with permission from [77]. (**D**) Magnetic-field-assisted gelation with two types—(ii) magnetic field or (iv) 3D printing—creates anisotropic alignment of magnetic nanoparticles. (**E**) Aligned structure promotes directional stem cell migration and elongation (white arrow: direction of applied magnetic field; yellow arrow: orientation of aligned iron oxide filaments). Scale bar = 200 µm. (**F**) Stem cells cultured within the aligned matrix by magnetic field (yellow arrow) exhibit enhanced myogenic differentiation, suggesting the role of topographic guidance in cell fate regulation. Panels (**i**–**vi**) illustrate the immunofluorescence staining of myogenic differentiation markers, demonstrating enhanced expression and organization of myotube structures in stem cells cultured within aligned magnetic hydrogel matrices. Reproduced with permission from [65].

**Figure 4 ijms-26-04987-f004:**
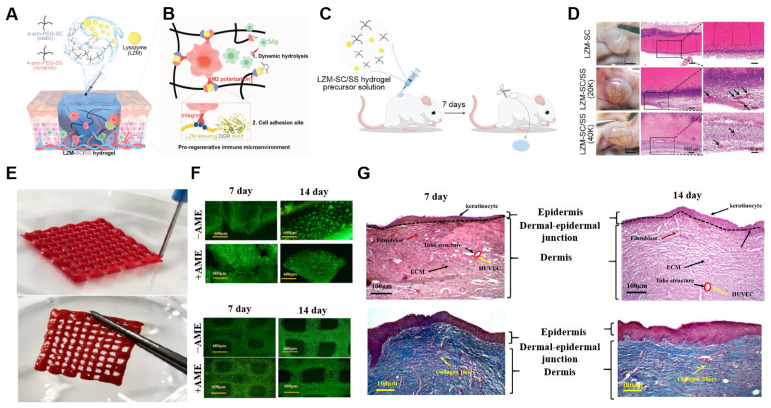
Application of ECM-mimetic hydrogels on tissue engineering. (**A**) Schematic illustration of the composition of the LZM-SC/SS hydrogel. (**B**) Mechanism of immune modulation by ECM-mimetic hydrogel. (**C**) Subcutaneous injection of the LZM-SC/SS hydrogel. (**D**) Histological evidence of macrophage polarization and angiogenesis (black arrows) after injection of the hydrogel. Reproduced with permission from [92]. (**E**) Three-dimensional bioprinting of the bilayered GelMA/gelatin hydrogel scaffold. (**F**) Effect of AME on cell viability within the scaffold. (**G**) Histological assessment of epidermal–dermal stratification and ECM formation. Reproduced with permission from [93].

**Figure 5 ijms-26-04987-f005:**
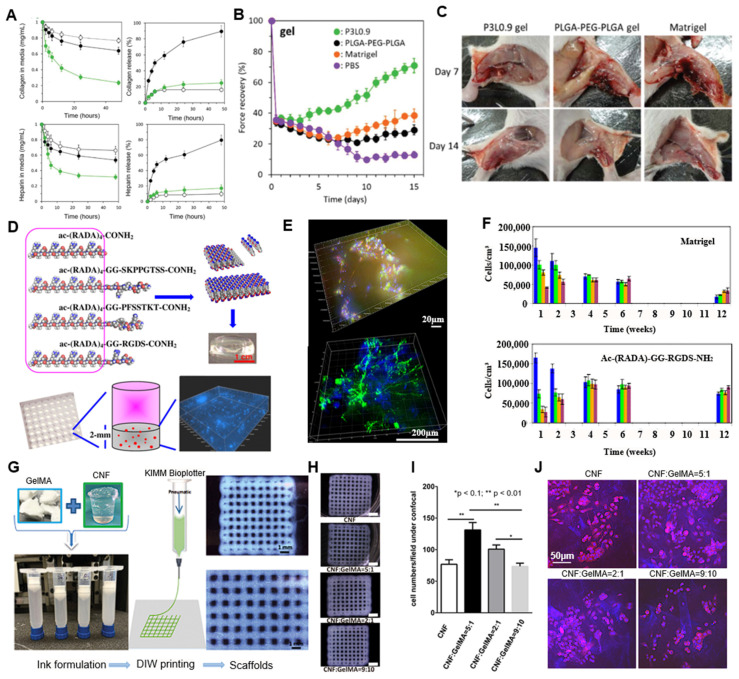
Application of ECM-mimetic hydrogel nanocomposite on tissue engineering. (**A**) Adsorption and release profiles of FITC-labeled ECM components (collagen and heparin) in PLGA-PEG-PLGA/LAPONITE^®^ ECM-mimetic hydrogel nanocomposite compared to control gels. (**B**) Quantitative analysis of muscle function recovery. Green: P3L0.9, White: Laponite, Black: PLGA-PEG-PLGA. (**C**) Macroscopic evaluation of muscle regeneration over time period in vivo. Reproduced with permission from [98]. (**D**) Schematic illustration for fabrication of self-assembling peptide hydrogel. (**E**) Three-dimensional confocal imaging of differentiated neural cells within the peptide hydrogel (top) 3D confocal imaging of differentiated neural cells within the peptide hydrogel (bottom). (**F**) Time-dependent differentiation profiles of neural stem cells encapsulated in functionalized peptide hydrogels. Colorcode: neuralprogenitorcells (blue), neurons (green), astrocytes (orange), and oligodendrocytes (brown). Reproduced with permission from [103]. (**G**) Formulation and 3D printing of CNF/GelMA ECM-mimetic hydrogel nanocomposite. (**H**) Effect of CNF/GelMA ratio on scaffold fidelity. (**I**) Mechanical analysis of printed CNF/GelMA scaffolds. (**J**) Cell viability and morphology within CNF/GelMA scaffolds. Reproduced from [104].

**Table 1 ijms-26-04987-t001:** Representative applications of ECM-mimetic hydrogels and ECM-mimetic hydrogel nanocomposites applied in cell culture modeling.

Field	Hydrogel Type	Nanomaterial Functionality	Hydrogel System	Target Tissue of Cell	Synthesis Method	Reference
Cell culture Modeling	ECM-mimetic Hydrogel	None	PEG-Laminin hydrogel	Neuroepithelial organoids	Ligand-functionalized polymer with enzymatic crosslinking	[60]
			PEG-Maleimide hydrogel	Intestinal organoids	Enzymatic and in situ crosslinking with biofunctionalization	[61,62,63,64,65]
			PEG-RGD hydrogel	iPSC-derived fibroblasts	Photo-crosslinkable synthetic polymer with adhesion motifs	[45,65,66]
	ECM-mimetic Hydrogel Nanocomposite	Electromagnetic properties	Hyaluronic acid/Alginate + Ti_3_C_2_ MXene	Neural tissue	Physical mixing and photocrosslinking	[67]
			Magnetic hydrogel + gold NPs + filamentous phage	3D cell culture platform	Magnetic labeling and levitation-based assembly	[68]
		Stimuli responsiveness	GelMA + Iron oxide nanoparticles	Skeletal muscle (C2C12)	Field-assisted alignment during photocrosslinking	[65]
			Hyaluronic acid + SPIONs	Neuronal cells	Microparticle incorporation with UV crosslinking	[69]
Tissue Engineering	ECM-mimetic Hydrogel	None	RGD peptide	Skeletal muscle (satellite cells)	Injectable chemically modified hydrogel (self-setting)	[70,71]
	ECM-mimetic Hydrogel Nanocomposite	Electromagnetic properties	GelMA + Gold nanorods	Cardiac tissue	Nanoparticle dispersion followed by UV curing	[59]
			GelMA + Graphene nanoplatelets	Neural tissue scaffold	3D bioprinting with photopolymerizable matrix	[72]
		Mechanical reinforcement	ChiMA + Nanosilicate clay	Bone tissue scaffold	3D printing of nanocomposite followed by UV curing	[73]
			GelMA + CNF	Cartilage (chondrocytes/iPSC)	Blending and extrusion-based 3D bioprinting	[74,75,76]
		Stimuli responsiveness	GelMA + CNTs	Cardiomyocytes	Nanomaterial dispersion and UV photocrosslinking	[77]
			Magnetic fibrin + γ-Fe_2_O_3_ NPs conjugated with bFGF/NGF/GDNF	Nasal olfactory mucosa (NOM)	In situ enzymatic gelation with pre-conjugated nanoparticle	[78]

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
