# Peer review of "Extracellular-Matrix-Mimetic Hydrogels by Using Nanomaterials"

_ijms, 2025, doi:10.3390/ijms26114987_

Round 1
Reviewer 1 Report
Comments and Suggestions for Authors
In this review, the authors provide a brief summary of the extracellular matrix (ECM) and the development of biomimetic hydrogels, with a particular focus on nanocomposite hydrogels. The main concerns raised by the reviewers are as follows:
- The classification of biomimetic hydrogels into synthetic hydrogels and nanocomposite hydrogels is somewhat inappropriate. For instance, synthetic hydrogels typically refer to those not derived from natural sources, in contrast to natural hydrogels, rather than serving as a distinct category of biomimetic materials.
- The incorporation of nanomaterials into hydrogels—resulting in nanocomposite hydrogels—is often intended not to mimic the ECM, but to enhance functional properties such as electrical conductivity, magnetic responsiveness, and more. Therefore, it may be more appropriate to classify them as functional hydrogels.
- Numerous important types of nanocomposite hydrogels and many representative studies or reviews have been reported in the literature, but are missing from the current review. For examples,
- Huang, G. et al. Functional and biomimetic materials for engineering of the three-dimensional cell microenvironment. Chem. Rev. 117, 12764-12850 (2017).
- Li, Y. H. et al. Magnetic hydrogels and their potential biomedical applications. Adv. Funct. Mater. 23, 660-672 (2013).
- Li, T. et al. Robust and sensitive conductive nanocomposite hydrogel with bridge cross-linking–dominated hierarchical structural design. Science Advances 10, eadk6643 (2024).
- …
- The quality of the figures needs to be significantly improved.
- Capitalization rules should be applied consistently across both figures and text.
- The full term "extracellular matrix (ECM)" only needs to be introduced at its first mention.
Reviewer 2 Report
Comments and Suggestions for Authors
The authors have submitted a review article, which mainly focus on the evolution from Matrigel to synthetic extracellular matrix (ECM)-mimetic hydrogel and nanocomposite ECM-mimetic hydrogels, and analyze the advantages and disadvantages of the transition from Matrigel, synthetic hydrogels, to nanocomposite hydrogels. The topic is meaningful as the nanocomposite ECM-mimetic hydrogels is an ideal candidate for 3D cell culture and tissue engineering. However, some issues should be carefully concerned.
- Figure 2-5, the number is in capital letters (A, B, C), but it’s in lowercase in the main text (a, b, c), please be uniform.
- Line 165, ‘4. Materials and Methods’, delete.
- In Section 4, are there any disadvantages or concerns for ECM-mimetic hydrogels when using in cell culture modeling and tissue engineering? Especially in the introduced references.
- Outlook should be enriched in the Conclusion section, for example, how to overcome the disadvantages of nanocomposite hydrogel listed in Figure 1.
- The role/mechanism of nanocomposite to improve the performance of hydrogel should discuss in detail.
Reviewer 3 Report
Comments and Suggestions for Authors
Comments to authors:
This review explores the evolution of extracellular matrix (ECM)-mimetic hydrogels from Matrigel to synthetic and nanocomposite formulations. It highlights the limitations of Matrigel, such as batch variability and limited mechanical tunability, and describes how synthetic hydrogels offer improved reproducibility and biocompatibility. The incorporation of nanomaterials, including carbon nanotubes and magnetic nanoparticles, further enhances hydrogel functionality, such as conductivity and responsiveness. Applications in 3D cell culture, vascular models, and tissue engineering are emphasized. Examples demonstrate improved outcomes in cardiomyocyte alignment, skin regeneration, and neural tissue engineering. The authors advocate for rational integration of nanomaterials to achieve next-generation, multifunctional ECM-mimetic systems.
[Line 18] – The authors propose that ECM-mimetic hydrogels must transcend passive structural mimicry. Could the authors define the specific biological functions or dynamic properties that such next-generation hydrogels are expected to emulate?
[Line 43] – Considering the xenogeneic origin and ethical limitations of Matrigel, have the authors evaluated or referenced any regulatory-approved synthetic or nanocomposite ECM substitutes suitable for clinical translation?
[Line 54] – The review highlights hybrid hydrogels combining natural and synthetic polymers. What are the primary challenges in preserving mechanical consistency, biochemical uniformity, and scalability in such composite systems?
[Line 67] – The manuscript discusses nanomaterial-mediated dynamic control of cellular behavior. Can the authors elaborate on the responsiveness mechanisms (e.g., photothermal, magneto-mechanical) and provide comparative performance data?
[Line 102] – Given that Matrigel is derived from murine sarcoma, how does its tumorigenic origin influence cell fate decisions differently compared to synthetic or nanocomposite ECM analogs?
[Line 125] – Potential cytotoxicity is acknowledged for nanomaterials. Have dose-dependent cytotoxicity studies or long-term biocompatibility assessments been conducted for the nanocomposite hydrogels referenced?
[Line 164] – While Table 1 outlines key hydrogel examples, it lacks critical evaluation. Could the authors expand this section to include a comparative analysis of physicochemical characteristics, bioactivity, and clinical applicability?
[Line 221] – In the context of CNT-GelMA hydrogels, how was electrical conductivity measured, and was a direct correlation established between conductivity levels and cardiomyocyte electrophysiological maturation?
[Line 239] – Magnetic nanoparticles were shown to influence stem cell differentiation. Were control studies performed to exclude the effects of magnetic field-induced thermal heating or mechanical deformation?
[Line 270] – The manuscript discusses macrophage polarization in response to LZM-SC/SS hydrogels. How were M1/M2 phenotypes quantified (e.g., surface markers, cytokine profiling), and were any systemic immune responses evaluated?
[Line 321] – For RADA peptide-based neural scaffolds, was the specificity of incorporated peptide motifs validated through transcriptomic or proteomic analyses to confirm directed differentiation toward neuronal subtypes?
[Line 335] – The CNF/GelMA hydrogel is praised for improved printability and biocompatibility. Have the authors assessed long-term structural integrity, degradation kinetics, or mechanical fatigue under physiological conditions?
Round 2
Reviewer 1 Report
Comments and Suggestions for Authors
no further comments